# Expression Analysis of MaTGA8 Transcription Factor in Banana and Its Defence Functional Analysis by Overexpression in Arabidopsis

**DOI:** 10.3390/ijms22179344

**Published:** 2021-08-28

**Authors:** Ping Lin, Tao Dong, Wenliang Chen, Niexia Zou, Yinglong Chen, Yuqing Li, Kelin Chen, Mingyuan Wang, Jianfu Liu

**Affiliations:** 1Institute of Horticulture Science and Engineering, Huaqiao University, Xiamen 361021, China; L_ping0509@stu.hqu.edu.cn (P.L.); wenliangchen@stu.hqu.edu.cn (W.C.); 19014087041@stu.hqu.edu.cn (N.Z.); 18013087012@stu.hqu.edu.cn (Y.L.); kelin@stu.hqu.edu.cn (K.C.); 2Key Laboratory of South Subtropical Fruit Biology and Genetic Resource Utilization, Ministry of Agriculture, Key Laboratory of Tropical and Subtropical Fruit Tree Research of Guangdong Province, Institution of Fruit Tree Research, Guangdong Academy of Agricultural Sciences, Guangzhou 510640, China; taod2004@163.com; 3The UWA Institute of Agriculture, and School of Agriculture and Environment, The University of Western Australia, Perth 6009, WA, Australia; yinglong.chen@uwa.edu.au

**Keywords:** *MaTGA8* transcription factor, *MaNPR1*, *Fusarium oxysporum* f. sp. *cubense* race 4, salicylic acid

## Abstract

TGA transcription factor is a member of the D subfamily of the basic region-leucine zippers (bZIP) family. It is a type of transcription factor that was first identified in plants and is the main regulator in plant development and physiological processes, including morphogenesis and seed formation in response to abiotic and biotic stress and maintaining plant growth. The present study examined the sequence of the *MaTGA8* transcription factor, the sequence of which belonged to subfamily D of the bZIP and had multiple cis-acting elements such as the G-box, TCA-element, TGACG-element, and P-box. Quantitative real time polymerase chain reaction (qRT-PCR) analyses showed that *MaTGA8* was significantly down-regulated by the soil-borne fungus *Fusarium oxysporum* f. sp. *cubense* race 4 (Foc TR4). Under the induction of salicylic acid (SA), *MaTGA8* was down-regulated, while different members of the *MaNPR1* family responded significantly differently. Among them, *MaNPR11* and *MaNPR3* showed an overall upward trend, and the expression level of *MaNPR4*, *MaNPR8,* and *MaNPR13* was higher than other members. *MaTGA8* is a nuclear-localized transcription factor through strong interaction with *MaNPR11* or weaker interaction with *MaNPR4*, and it is implied that the *MaPR* gene can be activated. In addition, the *MaTGA8* transgenic *Arabidopsis* has obvious disease resistance and higher chlorophyll content than the wild-type *Arabidopsis* with the infection of Foc TR4. These results indicate that *MaTGA8* may enhance the resistance of bananas to Foc TR4 by interacting with *MaNPR11* or *MaNPR4*. This study provides a basis for further research on the application of banana TGA transcription factors in Foc TR4 stress and disease resistance and molecular breeding programs.

## 1. Introduction

The TGA transcription factor is in the D subfamily of the basic leucine zipper (bZIP) family and specifically recognizes and binds to activation sequence 1 (as-1) with TGACGT as the core [1], while the leucine zipper region participates in oligomerization. The N-terminus of the transcription factor contains the acidic activation region STDxDT that can form a dimer, and the basic region binds directly to DNA [2]. The transcription factor can activate or inhibit the transcriptional expression of downstream interaction genes to regulate the normal growth and development of plants under stress or non-stress. For example, ten TGA transcription factors have been isolated and identified from *Arabidopsis*, which can be divided into five categories; members of groups I, II, and III are widely involved in the disease resistance of plants [3,4,5], and members of groups IV and V play a role in floral organ development [6,7,8]. To date, the TGA transcription factor has been isolated and identified in many plants, including papaya, rice, soybean, and strawberry [9,10,11,12]. It has been confirmed that the TGA transcription factor is involved in the process of drought resistance, salt tolerance, and detoxification, and promotes root nitrate absorption [13,14,15,16,17].

Most TGAs in the D subfamily have the ability to improve plant resistance to pathogens and xenobiotic stress. The class II *TGA2*, *TGA5*, and *TGA6* perform as key transcriptional regulators in Systemic Acquired Resistance (SAR) [18]. As salicylic acid (SA) concentration in the cell increases under pathogen infection, the NPR1 complex is decomposed and translocates into the nucleus, interacts with the TGA factors to induce the transcription of the antimicrobial *PR* genes, and enhances general resistance [19]. *(At)NPR1* transgenic rice overexpressing an *rTGA2.1* mutant that could no longer bind DNA or endogenous *rTGA2.1* was silenced. Both types of transgenic rice displayed increased tolerance to *Xanthomonas oryzae* pv. *Oryzae* (Xoo) and had changed expression levels of *PR* genes [20]. In addition, W-box is the cis-element of the WRKY *PR1* gene promoter. Tobacco *TGA2.2* can interact with the *NtWRKY12* transcription factor to regulate SA-inducible *PR-1* gene expression [21]. In the coordinated defense response of jasmonic acid (JA) and ethylene (ETH) hormones, *PLANT DEFFENSIN 1.2*, (*PDF1.2*) was used as the disease resistance marker gene. Interestingly, JA/ETH induces the impaired expression of *PDF1.2* and b-CHI in the tga2/5/6 triple mutant and decreases necrotizing *Botrytis cinerea* sensitivity [22]. Octadecanoid-responsive arabidopsis (AP2/ERF59, *ORA59*) is a transactivator that works upstream of *PDF1.2*. Additionally, it is strongly inhibited by SA in wild-type plants, which reduces the concentration of the *PDF1.2* gene [4]. It shows that TGA regulates plant disease resistance by mediating *ORA59*. Studies have shown that TGA transcription factors are regulated by the cytokinin (CTK)-activated transcription factor activated transcription factor 2 (*ARR2*) to enhance resistance to *Pseudomonas syringae* Pst *DC3000* [23].

*Fusarium* wilt, also known as Panama disease, is one of the most serious banana diseases destroying banana plants and is caused by the soil-borne fungus *Fusarium oxysporum* f. sp. *cubense* (Foc). When banana plants are infected by Foc, *MaNPR1A* and *MaNPR1B* are up-regulated, and the disease-resistant variety ‘GCTCV-218′ has a higher and earlier expression of *MaNPR1A* and *PR1* genes after inoculation, thereby reducing the sensitivity to *Fusarium* wilt [24]. Studies have shown that multiple genes involved in SA biosynthesis and downstream signal transduction pathways are suppressed in susceptible varieties and are activated in response to disease-resistant varieties after being inoculated with Foc tropical race 4 (Foc TR4) [25,26]. Since susceptible varieties inhibit the expression of genes related to SA biosynthesis and SA content, the induced resistance to Foc TR4 may be SA-dependent systemic acquired resistance [27,28,29]. To date, the molecular mechanism of the TGA transcription factors regulating banana disease resistance through the SA signaling pathway is not yet clear.

Therefore, we hypothesize that the banana TGA transcription factor interacts with NPR1 in the SA signaling pathway to increase resistance to Foc TR4. In our study, we cloned and obtained *MaTGA8* and performed sequence analysis. The response of *MaTGA8* and *MaNPR1* to exogenous SA was further analyzed, and potential NPR1 interacting proteins were screened and verified through yeast plug hybridization. We also investigated the molecular functions of the overexpression of *MaTGA8* in response to Foc TR4 in *Arabidopsis* plants.

## 2. Results

### 2.1. Analysis of MaTGA8 Gene

A phylogenetic tree was constructed for MaTGA8 and the TGAs family of *Oryza sativa* L. and *Arabidopsis thaliana*. By comparing the transcription factors MaTGA8, *Arabidopsis* AtTGA, and *Oryza sativa* L. OsbZIP to the transcription factor family, the MaTGA8 and OsbZIP transcription factors were found to have the closest evolutionary relationship (Figure 1); among them, banana and rice were both monophytes.

The protein structure of MaTGA8 was analyzed. ORF of MaTGA8 contained 504 amino acids. Further analysis through protein multiple sequence alignment indicated that the MaTGA8 contained one bZIP domain and one DOG1 domain; part of the bZIP domain was composed of 14 amino acid residues as a conserved DNA binding site, and the other part was the leucine zipper region involved in the formation of dimers (Figure 2). This indicates that *MaTGA8* belongs to the bZIP transcription factor family Group D subfamily.

### 2.2. Promoter Analysis for Detection of Putative Cis-Elements

The prediction of the cis-elements found that growth and development of related cis-elements identified were MRE, the TCCC-motif, the TCT-motif, the GATA-motif, the CAT-box, the G-box, and the RY-element. MBS is a cis-element involved in drought stress. The presence of hormone related cis-elements such as the TCA-element, TGACG-motif, CGTCA-motif, GARE-motif, P-box, and ABRE in promoter region indicates the responsiveness of *MaTGA8* towards SA responsiveness, MeJA responsiveness, gibberellin responsiveness, and abscisic acid reactivity (Table 1).


### 2.3. Expression Level of MaTGA8 and MaNPR1 under Stress Treatments

The Expression level of the banana *MaTGA8* transcription factor gene in response to Foc TR4 was investigated by quantitative real-time PCR for susceptible varieties such as Williams as well as in disease-resistant varieties such as Nan Tianhuang. The results showed that the expression level of *MaTGA8* in susceptible and disease-resistant banana variety roots was rapidly induced and significantly down-regulated under Foc TR4 infection. The expression was severely inhibited compared to inhibition at day 0 (before inoculation) (Figure 3A).

To identity potential NPR1 interaction genes, we analyzed the response of the *MaTGA8* and *MaNPR1* genes to SA. *MaTGA8* was significantly up-regulated at 48 h in the Nan Tianhuang variety, but the overall trend in both the Nan Tianhuang and Williams varieties was induced to down-regulate expression (Figure 3B), while there were obvious differences in the expression patterns of the *MaNPR1* family members in both the Nan Tianhuang and Williams varieties. *MaNPR1* was quickly induced by the treatment of 1 mM SA and increased up to the highest level of 7.62-fold and 26.07-fold at 24 and 48 h, respectively, in roots of both varieties (Appendix A). *MaNPR2*, *MaNPR8,* and *MaNPR15* were significantly down-regulated in the Nan Tianhuang roots. The expression level of *MaNPR2* and *MaNPR8* increased up to the highest level at 48 h compared to the control. *MaNPR15* reached its highest level at 24 h (Appendix A). The expression level of *MaNPR3* and *MaNPR11* showed an overall upward trend, increasing to the highest level of 6.42-fold and 13-fold at 48 in Williams roots and to the highest level of 10.01-fold and 7.39-fold at 12 and 24 h in Nan Tianhuang roots, respectively (Appendix A). The expression level of *MaNPR4* increased to the highest level of 200.85 over 12 h in Nan Tianhuang roots (in comparison to that of the control), then declined until the end of the experiment. Similar to *MaNPR4*, *MaNPR7* had the same expression pattern in the Nan Tianhuang roots. The expression level of *MaNPR5* and *MaNPR14* both showed higher induced expression at 24 h in the Nan Tianhuang roots (Appendix A). *MaNPR9* and *MaNPR13* showed higher induced expression at 24 h in the Williams roots, while *MaNPR10* and *MaNPR12* increased to the highest expression level at 12 h in the Nan Tianhuang roots (Appendix A).

### 2.4. Yeast Two-Hybrid Assays

The full-length coding sequence of *MaTGA8* was ligated to the binding domain (BD) of GAL4 in the pGBKT7 vector and was co-expressed with the coding sequences of *MaNPR3MaNPR4* and *MaNPR11* fused to the GAL4 transcription activation domain (AD) of pGADT7 vector in yeast containing the AbA and x-gal reporter genes. The results show that the co-expression of MaTGA8 and MaNPR3 appeared as white colonies, while the co-expression of MaNPR4 and MaNPR11 appeared as blue colonies, and the co-expression of MaNPR4 appeared as a lighter blue (Figure 4). This shows that MaTGA8 had a weak interaction with MaNPR4 and a strong interaction with MaNPR11. These results imply that *MaTGA8* could provide the DNA binding ability required by *MaNPR1* to activate the *MaPR* gene.

### 2.5. Sub-Cellular Localization of MusaTGA8

To determine the subcellular location of *MaTGA8*, the full-length coding sequence was amplified via PCR and was then fused to GFP driven by the cauliflower mosaic virus (CaMV) 35S promoter. The 35S:MaTGA8-GFP vectors were then transformed into onion epidermal cells via *A. tumefaciens* strain GV3101, and the Laser scanning confocal microscope was used to image this phenomenon. As shown in Figure 5, the empty vector 35S: GFP was transiently expressed throughout the nucleus and cytoplasm, while the 35S: MaTGA8-GFP fusion proteins were only located in the nucleus.

### 2.6. Overexpression of MaTGA8 Increase Foc TR4 Resistance in Arabidopsis

To determine the expression level of *MaTGA8* gene in transgenic lines, the transgenic plants were analyzed using RT-qPCR. We detected six *MaTGA8*-overexpression lines (tga8 OE-1~6), and the expression level of *MaTGA8* was up-regulated in transgenic *Arabidopsis*, compared to wild-type (WT) (Figure 6A). Among them, the expression level of the *MaTGA8* gene in OE-4 was 7.89 times that of the WT, and OE-3, with the lowest expression level, was 2.95 times that of the WT (Figure 6A). This shows that the *MaTGA8* gene was successfully transformed into *Arabidopsis* and that the mRNA was transcribed.

To test whether the reduced disease symptoms in *MaTGA8* transgenic plants was correlated with Foc TR4 multiplication, we checked for pathogen growth after Foc TR4 inoculation. For further analysis, the Foc TR4 qPCR detection system constructed by our laboratory was used to detect the abundance of Foc TR4 in *Arabidopsis*. The abundance of Foc TR4 in tga8 and WT showed an upward trend and reached 1.55 × 10^6^ (copies/mL) and 1.66 × 10^6^ (copies/mL) at 92 h, respectively. This showed that Foc TR4 successfully infected *Arabidopsis* (Figure 6B). With the passage of transgenic *Arabidopsis* infection time, the mRNA level of *MaTGA8* was decreased, while the overall expression level showed an upward trend (Figure 6C). Therefore, we speculated that the plant did not respond early when it was stressed by Foc TR4 and quickly generated a defensive response after recognition.

After three weeks, the Col-0 + Foc TR4 group and the tga8 + Foc TR4 group *Arabidopsis* plants were smaller than the Col-0 plants, and the plants produced different degrees of yellowing symptoms in the leaves, while the yellowing symptoms of the leaves in the Col-0 + Foc TR4 group were more serious than that of the tga8+ Foc TR4 group (Figure 6E). The chlorophyll content of the Col-0 + Foc TR4 group was significantly different from that of the tga8 + Foc TR4 group and Col-0 (Figure 6D). This illustrates that the overexpression the *MaTGA8* gene in *Arabidopsis* can enhance disease resistance to Foc TR4.

## 3. Discussion

The use of resistant varieties is the most effective means to manage this disease. The resistance of Cavendish to Fusarium wilt in banana Race 1 (R1) has had an enormous impact on the banana industry, contributing to near complete Cavendish dominance in export trade, as reported by Dita et al. [30]. To date, there are no commercial cultivars that are resistant to Fusarium wilt in banana Tropical Race 4 (TR4) with similar levels of resistance from Cavendish to Fusarium wilt in banana R1; hence, the importance and benefit of considering the presence of Fusarium wilt in banana and favorable environmental conditions such as soil, climate, management agronomic, among others [31].

In the present study, a TGA transcription factor, *MaTGA8*, from banana was identified, and its role in the regulation of Foc TR4 tolerance in *Arabidopsis* plants was analyzed. *MaTGA8* belongs to group-D of the bZIP transcription factor known to contain the bZIP-motif and the DOG1-motif. The plant bZIP transcription factor falls into 13 categories, and different subfamilies are involved in regulating different metabolic pathways, while transcription factors located in group D can participate in plant disease defense and physiological growth [32]. There is a scarcity of reports dealing with the functional analysis of the banana *TGA* transcription factor and their roles in regulation of defense responses. However, based on the available information, 121 bZIP transcription factor genes were identified in the banana genome and were classified into 11 subfamilies [33]. In the phylogenetic tree, MaTGA8 has a close genetic relationship with rice OsbZIP. *OsbZIP79* was reported as a negative regulator of diterpenoid phytoalexin production in rice cells [34]. Studies have shown that the cis-elements in the promoter region of a gene are closely related to the biological function of the gene [35]. Our analysis of the cis-acting elements showed that the promoter region of *MaTGA8* contained the pathogen-related cis-elements, such as the P-box or GARE-motif, indicating that *MaTGA8* is involved in the defense of pathogens. Therefore, we hypothesized that *MaTGA8* regulates plant immunity against Foc TR4.

Non-expression of pathogenesis-related gene 1 (NPR1) is a necessary positive regulator for SA-induced pathogenesis-related (PR) gene expression and plant SAR. NPR1 function is reduced from an oligomeric state to a monomeric state and is then translocated to the nucleus after the accumulation of SA induced by pathogen infection [36]. Importantly, class II TGA and several other TGAs physically interact with NPR1, which represents the main regulator of the plant pathogen response [3,18]. TGAs and NPR1 have been fully documented as two critical components to enhance SA-dependent PR gene transcription [37]. As a set of basic region/leucine zipper motif (bZIP) factors, TGA can interact with the transcription coactivator NPR1 and can provide its DNA binding ability to activate the representative SAR marker PR gene in plants [38]. Furthermore, our results of the qRT-PCR analysis showed that the expression level of MaNPR1 family members varied greatly in Willams and Nan Tianhuang varieties under exogenous SA (Appendix A). The application of MaNPR3, MaNPR4, and MaNPR11, the proteins of which are extremely homologous to *Arabidopsis* NPR1 as prey in Y2H analysis, and the MaNPR4 and MaNPR11 proteins could effectively interact with the MaTGA8 in yeast in vivo (Figure 4). This conclusion is similar to the results of other studies [23,39,40]. Mutations of the NPR1 gene (*SlNPR1*) failed to increase the resistance of tomato plants to *Pseudomonas syringae*, while the significant activation of SAR was recorded when the NPR1 gene was overexpressed [41].

In our study, the enhanced Foc TR4 resistance by *MaTGA8* was confirmed by its heterologous overexpression in transgenic *Arabidopsis* plants. Specifically, wild-type *Arabidopsis* and transgenic *Arabidopsis* under Foc TR4 infection conditions showed significantly weaker growth than uninfected wild-type *Arabidopsis*, while wild-type *Arabidopsis* plants showed more leaf wilting and lower chlorophyll content than transgenic *Arabidopsis* plants (Figure 6). Among rice bZIP transcription factors, *OsTGA2* could interact with rice NPR1-homolog proteins (NHs). Overexpression of *OsTGA2* increased the resistance of rice to bacterial leaf blight [10]. Studies have shown that *VvTGA1* can provide the DNA binding capacity required by *VvNPR1* to activate the *VvPR* gene, and the interaction between *VvNPR1* and *VvTGA1* may be the cause of the effective disease resistance induced by β-aminobutyric acid (BABA) [42]. The data show that the banana *MaTGA8* transcription factor interacts with *MaNPR4* or *MaNPR11* to activate the expression of the *MaPR1* gene downstream of the SA signaling pathway and promotes the disease resistance of bananas (Figure 7).

Previously, the *AtTGA1*-*AtTGA7* of *Arabidopsis* plants were divided into three clades related to plant defense based on sequence homology [43]. Class II TGAs can interact with glutaredoxin GRX480 and can mediate the inhibitory effect of SA on the JA signaling pathway marker gene *PDF1.2*. Meanwhile, Class II TGAs can also interact with the GRAS family protein SCL14 [2,4,44]. This interaction promotes the binding of TGA to the downstream *PR1* promoter, improves up-regulated expression, strengthens plant disease resistance. The SA signal pathway is administered with cytokinin, which regulates the combination of TGA3 and PR-1 promoter through ARR2 to improve plant disease resistance [23]. These studies indicate that the TGA transcription factors can mediate different signal pathways to improve plant disease resistance. However, in our study, only the SA signal pathway was discussed to regulate the process of resistance to Fusarium wilt in bananas. Whether other pathways regulate the process of Fusarium wilt has not been explained.

Our results could serve as the basis for future lines of research by molecular plant sciences pioneers in Colombia and Peru where Fusarium wilt in bananas (TR4) has recently been located according to Olivares et al. [45]; This would also be an essential contribution in adjacent banana areas of Latin America according to Olivares [46] in order to avoid the spread and devastation of banana plantations.

## 4. Materials and Methods

### 4.1. Plant Materials and Growth Conditions

There were two banana varieties, the Nan Tianhuang and Williams (Guangdong Shishengyuan Biological Technology Development Co., Ltd., Guangdong, China), that were used in this study. Tissue culture seedlings of the bananas were grown in nutrient soil and vermiculite mixture (3:1) in a greenhouse at 30 ± 1 °C. The young roots of banana with the first 5-leaf compound leaf unfolded were selected as the experimental materials and were treated with *Fusarium oxysporum* f. sp. *cubense* race 4 (Foc TR4) and salicylic acid (1 mM).

The wild-type (WT) *Arabidopsis* Col-0 seeds were sprinkled on a nutrient substrate (soil: vermiculite 2:1 *v/v*). *Arabidopsis* seedlings were transplanted to new nursery pots one week after sowing and continued to cultivate in the greenhouse (12 h light/dark cycles and Natural light at 24 °C); nutrient solution was supplied once a week.

### 4.2. Bacterial Strains

The Foc TR4 was inoculated on a potato dextrose agar (PDA, Solarbio, Beijing, China) medium for 5–7 d at 28 °C and was then transferred to liquid PDA medium for 3–5 d at 28 °C and 200 rpm shaking culture. The Foc TR4 solution was filtered by sterilized six-layer gauze to remove the mycelium, and the spore solution concentration was adjusted to 5 × 10^7^ spores/mL.

### 4.3. The Total RNA Extraction and Quantitative Real-Time PCR (qRT-PCR)

The plant RNA isolation was performed using a polysaccharide polyphenol plant RNA extraction kit (DP441, TIANGEN, Beijing, China). The total RNA integrity was confirmed using 1% agarose gel electrophoresis. RNA concentration and quantity were measured with the microspectrophotometer ND2000C (ThermoFisher, Waltham, MA, USA). Then, the total RNA was used for cDNA synthesis using a PrimeScript™ 1st Strand cDNA Synthesis Kit following the manufacturer’s protocol (TaKaRa, Japan). For the qRT-PCR, each microcentrifuge tube contained 2 μL of diluted cDNA, 10 μL of TB Green Premix (TaKaRa, Japan), 1 μL of gene-specific primers (10 mM), and 6 μL ddH_2_O. The PCR amplification procedure was as follows: predenaturation at 95 °C for 30 s followed by 40 cycles of 95 °C for 5 s, 55 °C for 30 s, and 72 °C for 30 s. Finally, the dissociation melting curve program was set up. The specificity of the PCR products was confirmed by dissociation melting curve analysis. The housekeeping genes *MaActin* and *AtActin* were used as internal references for banana and *Arabidopsis*, respectively. Each sample test was set to produce three biological replications. The values of the transcript abundance were calculated using the 2^−^^△△Ct^ method. All of the primer sequences used in this study are listed in Appendix A.

### 4.4. Cloning and Sequence Analysis of MaTGA8

The *MaTGA8* gene (ID: Ma11_g13030) was amplified from banana cDNA. The cloning specific primer is described in Appendix A. The PCR amplification system was as follows: 1μL each of forward primer and reverse primer (10 mM), 2 μL of diluted cDNA, 2.5 μL of 10 × buffer, 2 μL of dNTP Mix (2.5 mmol/L), 0.25 of LA polymerase (TaRaKa, Japan), and ddH_2_O were mixed to a volume of 25 μL. The PCR running program was 94 °C for 3 min followed by 35 cycles of 94 °C for 30 s, 56 °C for 30 s, 72 °C for 5 min, and a final extension of 72 °C for 10 min. The coding sequences of MaTGA8 were cloned into the pMD-18-T vector (TaRaKa, Japan) and were then transformed into *E. coli* DH5α (Weidi Biotechnology Co., Ltd., Shanghai, China) for verification.

The sequences of *MaTGA8* were used to construct a phylogenetic tree using MEGA 7.0 software (Mega Limited, Auckland, New Zealand) with the neighbor-joining method (with boot strap replications of 1000). The MaTGA8, rice, and *Arabidopsis* TGA proteins were aligned using multi-sequence alignment by means of ClustX (http://www.clustal.org/clustal2/#Webservers accessed on 5 April 2020). To understand the role of the *MaTGA8* gene in the growth and development of banana plants, we predicted the cis-acting elements of *MaTGA8* using the PlantCare database (http://bioinformatics.psb.ugent.be/webtools/plantcare/html/ accessed on 2 May 2020).

### 4.5. Plastid Construction and Generation of Transgenic Arabidopsis Plants

According to the restriction endonuclease sites on the expression maps of the pMD-18-T and pCAMBIA-1302 vectors, two restriction endonucleases, *Pst* I and *Kpn* I (TaRaKa, Japan), were individually used to double digeste the pMD-MaTGA8 and pCAMBIA-1302 vectors. Additionally, then the full-length coding sequence of *MaTGA8* was inserted between the *Pst* I and *Kpn* I sites of the pCAMBIA-1302 vector.

To obtain stable transgenic *Arabidopsis* lines, the recombinant expression vector pCAMBIA-1302-MaTGA8 was transformed into the *Agrobacterium tumefaciens* strain GV3101 (Weidi Biotechnology Co., Ltd., Shanghai, China) and was then transformed into *Arabidopsis* ecotype Col-0 by means of the floral dip method [47]. The resultant T0 seedlings were screened on resistant media that contained the hygromycin and were confirmed via qRT-PCR. T3 homozygous transgenic lines were used for experiments.

### 4.6. Subcellular Localization of MaTGA8

The overexpression of the pCAMBIA-1302 vector containing 35S: MaTGA8-GFP was used to conduct the subcellular localization experiments. This expression vector was transformed into the *Agrobacterium tumefaciens* GV3101 strain. Square (1 cm) onion epidermal cells were cut from sterilized fresh onion bulbs and were immediately submerged into a suspension of *A. tumefaciens* at OD600 = 1. After inoculation for 20 min, infected onion epidermal cells were used with filter papers to absorb excess agrobacterium solution and were coincubated on 1/2 MS medium for 3 days at 25 °C. A 35S: GFP blank vector was used as a positive control. GFP imaging was performed under a laser confocal microscope (LEICA TCS SP8).

### 4.7. Yeast Two-Hybrid Assays

Two-hybrid (Y2H) screening was performed according to the manufacturer’s protocols using the Matchmaker™ Gold Yeast Two-Hybrid System (Clontech, TaRaKa, Japan). The full-length sequence of *MaTGA8* was cloned into the bait vector pGBKT7 and was then transformed into the yeast strain Y2HGold. X-a-Gal and aureobasidin A (AbA) were used as a yeast selection marker and were added to the selection medium to reduce the autonomous activation of pGBKT7-MaTGA8. To confirm *MaTGA8* and the three MaNPR1 interactions, the full-length *MaNPR3*, *MaNPR4,* and *MaNPR11* was cloned into the prey vector pGADT7. Additionally, the pGADT7-MaNPR1 plasmid was then transferred to Y2HGold (containing pGBKT7-TGA8) Yeast competent by means of a co-transformation method; the positive cloned strains were cultured on a SD-Ade-His-Leu-Trp-X-a-gal-AbA solid plate.

### 4.8. Foc TR4 Treatment in Transgenic Arabidopsis

The transgenic *Arabidopsis* with the overexpression of *MaTGA8* and the Col-0 control group were selected to conduct the Foc TR4 tolerance assays. Transgenic *Arabidopsis* (tga8) and Col-0 were irrigated with 30 mL Foc TR4 (the tga8 + Foc TR4 and Col-0 + Foc TR4 group) after the third week of transplantation. The phenotypic differences between the transgenic *Arabidopsis* and Col-0 were then observed after three weeks of treatment. The expression level of *MaTGA8* in transgenic *Arabidopsis* and the abundance of Foc TR4 in *Arabidopsis* were also measured at the initial stage of Foc TR4 treatment.

### 4.9. Statistical Analysis

In this study, there were at least three biological replicates for all experiments. All data were analyzed using ANOVA and Student’s *t*-test and were expressed as means ± standard deviation (SD). The lowercase letters (a, b …) and asterisk (*) were used to indicate the differences between the two analysis methods at *p* < 0.05.

## 5. Conclusions

The *MaTGA8* gene was characterized as a nuclear localization transcription factor and belongs to the D subfamily of the bZIP family. The Foc TR4 infection triggered a defense response that was intently related to the SA-mediated immune regulatory pathway. *MaNPR4* and *MaNPR11* were confirmed to be the downstream target of *MaTGA8* by direct protein–protein interaction. In addition, overexpression of *MaTGA8* significantly increased the resistance of transgenic *Arabidopsis* to Foc TR4 and slowed the growth and development of transgenic *Arabidopsis*. In summary, this study has shown that *MaTGA8* can provide the DNA binding capacity required by *MaNPR4* and *MaNPR11* to activate the *MaPR1* gene and can improve the disease resistance of bananas by regulating the SA-mediated signal pathway. This means that *MaTGA8* may jointly regulate banana resistance through other plant hormone signaling pathways.

## Figures and Tables

**Figure 1 ijms-22-09344-f001:**
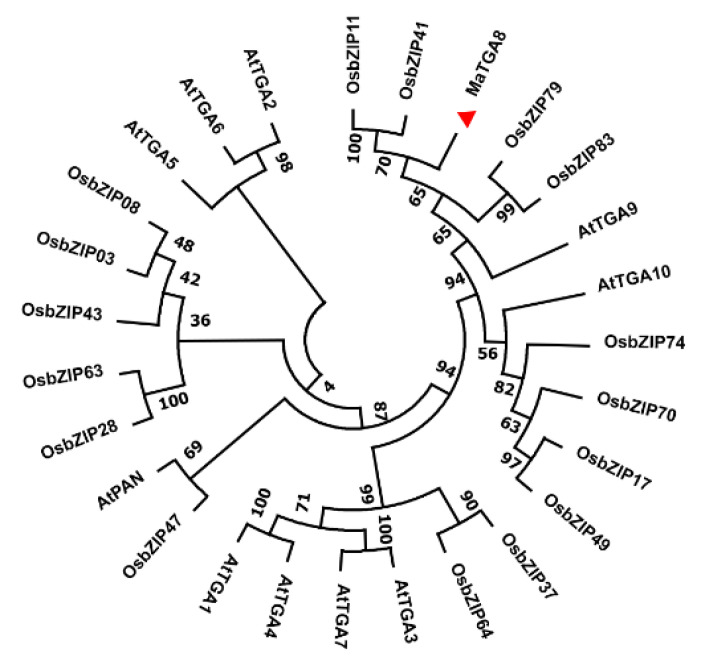
Phylogenetic analysis of *MaTGA8* and the TGAs family of *Aarabidopsis* and *Oryza sativa*.

**Figure 2 ijms-22-09344-f002:**
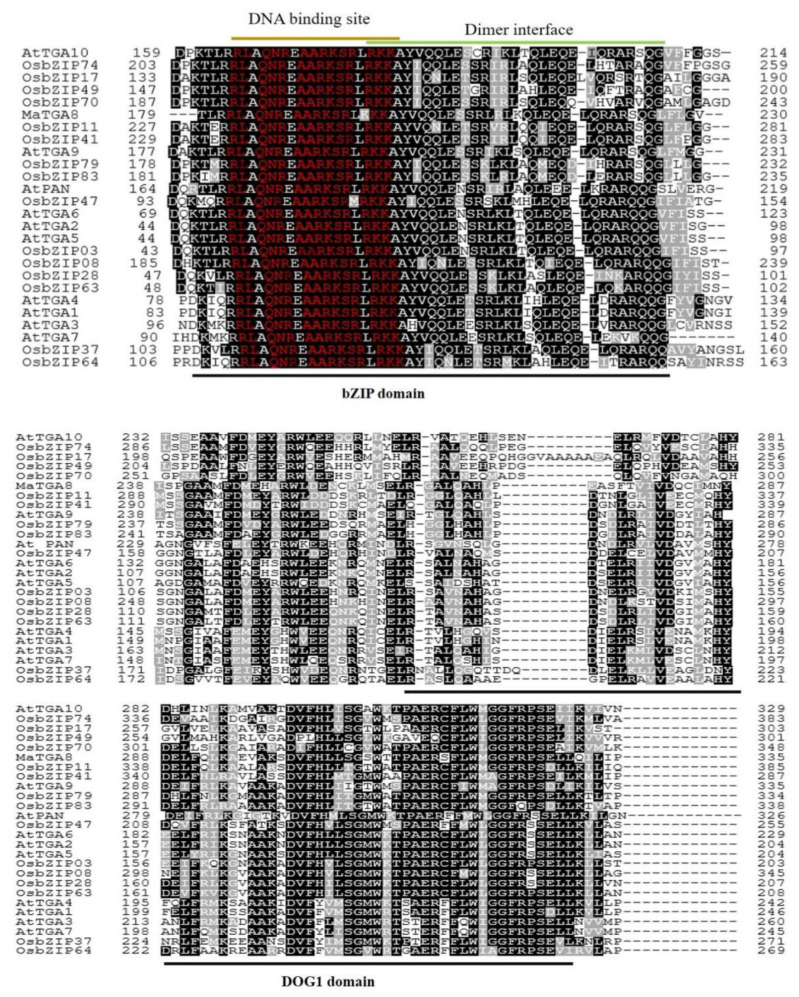
The protein sequence alignment of *MaTGA8* and the TGAs family of *Aarabidopsis* and *Oryza sativa*. The sequences marked with thick black line are the conserved bZIP domain and the DGO1 domain. The sequences with a black background indicate completely identical residues, and the sequences with a gray background indicate similar residues.

**Figure 3 ijms-22-09344-f003:**
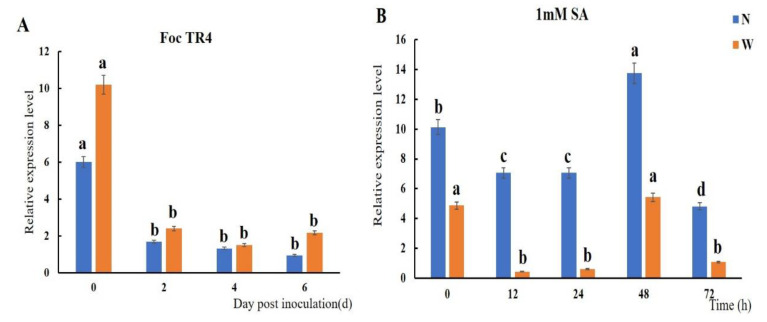
The expression level of *MaTGA8* under Foc TR4 (**A**) and SA (**B**) treatment. N: Nan Tianhuang, W: Williams. Symbols a, b, c, etc., indicate a significant difference in the *p* values of 0.05.

**Figure 4 ijms-22-09344-f004:**
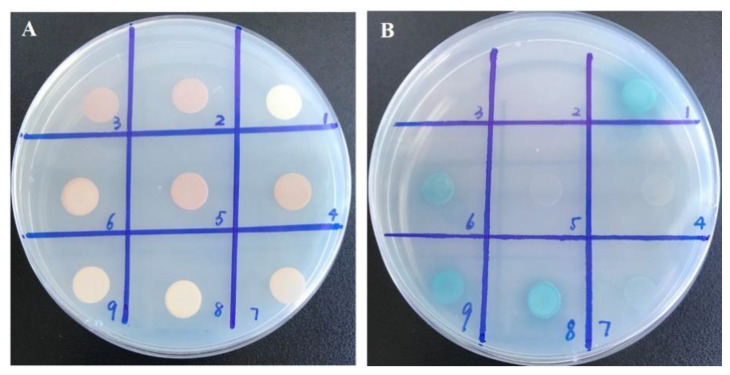
MaTGA8 interacts with MaNPR3, MaNPR4, and MaNPR11. Control (**A**); mutual verification (**B**); 3, 4, 5, and 6: auto-activation of pGADT7-NPR3, pGADT7-NPR4, pGADT7-NPR11, and pGBKT7-TGA8 were detected. 7, 8, 9: pGADT7-NPR3, pGADT7-NPR4, and pGADT7-NPR11 interaction with pGBKT7-TGA8 were detected. Growth of transformed yeast was evaluated on SD/-Ade/-His/-Leu/-Trp/X-a-gal/Aba medium.

**Figure 5 ijms-22-09344-f005:**
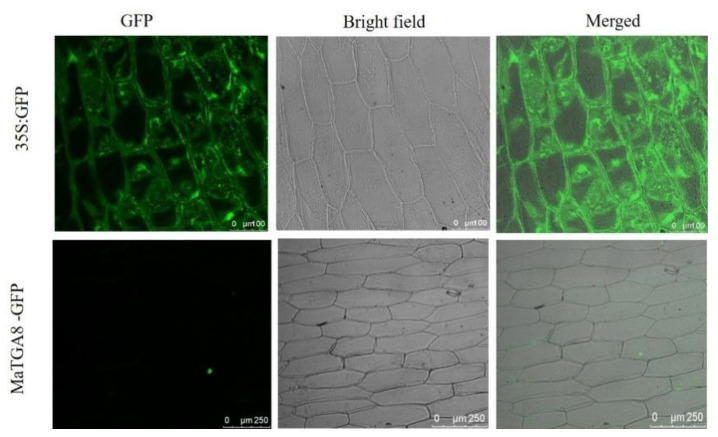
Subcellular localization of MaTGA8 gene.

**Figure 6 ijms-22-09344-f006:**
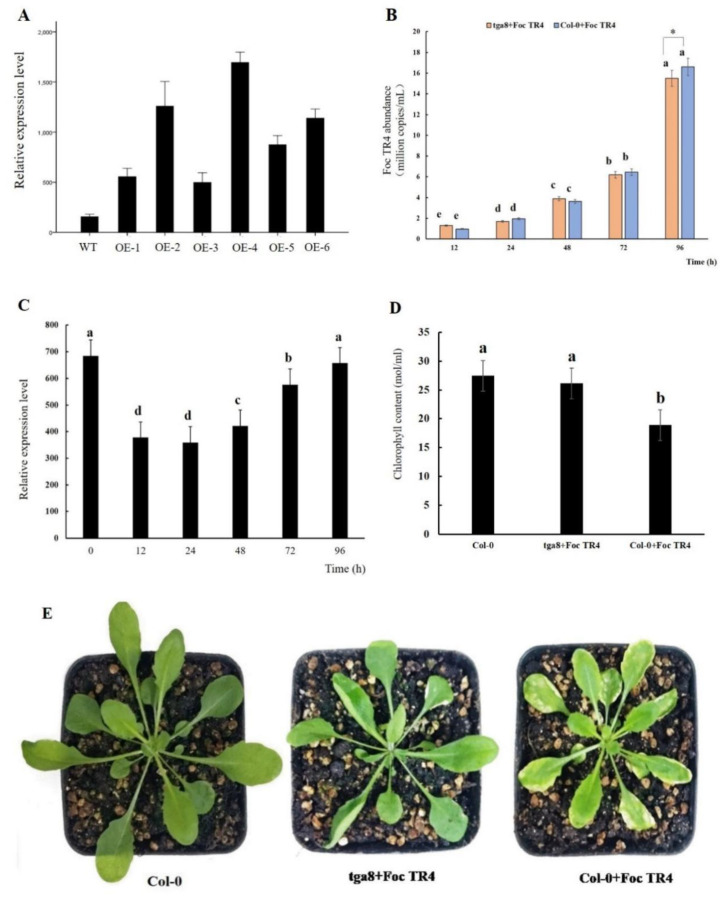
Improved Foc TR4 resistance in the transgenic *Arabidopsis* plants with the heterologous overexpression of *MaTGA8*. The relative expression of transgenic *MaTGA8 Arabidopsis* in six randomly chosen lines (**A**). Foc TR4 abundance in *Arabidopsis* plants (**B**). Expression level of *MaTGA8* in transgenic *Arabidopsis* after FocTR4 infection (**C**). *Arabidopsis* chlorophyll content (**D**). Phenotypes of transgenic *Arabidopsis* and wild-type Arabidopsis after Foc TR4 infection (**E**). Symbols a, b, c, etc., indicate a significant difference in the *p* values of 0.05.

**Figure 7 ijms-22-09344-f007:**
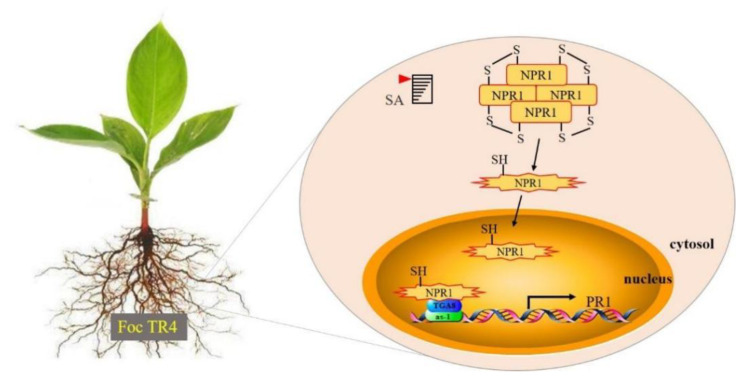
Working model of MaTGA8 regulates bananas to improve disease. Bananas are invaded by Foc TR4, SA accumulates in the plant, and NPR1(MaNPR4/MaNPR11) is reduced from an oligomeric state to a monomeric state and is then translocated to the nucleus, interacts with TGA factors, and promotes the expression of PR genes.

**Table 1 ijms-22-09344-t001:** Prediction of the cis-acting elements of the MaTGA8 gene.

Motif	Function
MRE	Light-response element
TCCC-motif	Light-response element
TCT-motif	Light-response element
G-box	Light-response element
GATA-motif	Light-response element
GT1-motif	Light-response element
CAT-box	Meristem expression
TCA-element	Involved in SA responsiveness
TGACG-motif	MeJA response element
CGTCA-motif	MeJA response elements
GARE-motif	Gibberellin response element
P-box	Gibberellin response element
ABRE	Abscisic acid reactive cis-acting element
MBS	MYB binding site involved in drought-inducibility
RY-element	Seed-specific regulation

## Data Availability

Not applicable.

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
