# Peer review of "Expression Analysis of MaTGA8 Transcription Factor in Banana and Its Defence Functional Analysis by Overexpression in Arabidopsis"

_ijms, 2021, doi:10.3390/ijms22179344_

Round 1
Reviewer 1 Report
Your work is interested, written well, and organized. However, there are some comments should be considered before publishing. However, it is necessary to focus a couple of final paragraphs in the discussion that would improve the scientific quality of the manuscript.
I continue to add a paragraph that summarizes the importance, usefulness and social relevance, contemporary of the study, specifically pointing out the Impact, Benefit and Social Projection, something like this:
…The use of resistant varieties is the most effective means to manage this disease. The resistance of Cavendish to Fusarium wilt in banana Race1 (R1) has had an enormous impact on the banana industry contributing to near complete Cavendish dominance in export trade as reported by Dita et al. (2018) [Add]. To date, there are no commercial cultivars resistant to Fusarium wilt in banana Tropical Race 4 (TR4) with similar levels of resistance from Cavendish to Fusarium wilt in banana R1, hence the importance and benefit of considering the presence of Fusarium wilt in banana and favorable environmental conditions such as soil, climate, management. agronomic, among others (Olivares et al. 2020) [Add].
… Our results could serve as the basis for future lines of research by molecular plant sciences pioneers in Colombia and Peru where Fusarium wilt in bananas (TR4) is recently located according to Olivares et al. (2021a) [Add], it would also be an essential contribution in adjacent banana areas of Latin America according to Olivares (2021b) [Add] to avoid the spread and devastation of banana plantations.
Dita, M.; Barquero, M.; Heck, D.; Mizubuti, E.; Staver, C.P. Fusarium Wilt of Banana: Current Knowledge on Epidemiology and Research Needs Toward Sustainable Disease Management. Frontiers in Plant Science, 2018, 9,1468. https://doi.org/10.3389/fpls.2018.01468
Olivares, B.O., Araya-Alman, M., Acevedo-Opazo, C. et al. Relationship Between Soil Properties and Banana Productivity in the Two Main Cultivation Areas in Venezuela. Journal of Soil Science and Plant Nutrition, 2020, 20, 2512–2524 https://doi.org/10.1007/s42729-020-00317-8
Olivares, B.O.; Rey, J.C.; Lobo, D.; Navas-Cortés, J.A.; Gómez, J.A.; Landa, B.B. Fusarium Wilt of Bananas: A Review of Agro-Environmental Factors in the Venezuelan Production System Affecting Its Development. Agronomy, 2021, 11, 986. https://doi.org/10.3390/agronomy11050986
Olivares Campos, B.; Paredes, F.; Rey, J.; Lobo, D.; Galvis-Causil, S. The relationship between the normalized difference vegetation index, rainfall, and potential evapotranspiration in a banana plantation of Venezuela. SAINS TANAH - Journal of Soil Science and Agroclimatology, 2021, 18,1, 58-64. http://dx.doi.org/10.20961/stjssa.v18i1.50379
Reviewer 2 Report
In this manuscript the isolation and characterization of banana MaTGA8 gene and its role in defence responses against Fusarium oxysporum f. sp. Cubense are described. Although the topic is interesting and several very nice results are presented, in my opinion this paper is not suitable for publication in IJMS. My main concern is that, generally, it is poorly written. Either its scientific and English languages and style are not sufficient, thus it should be re-written.
Some critical remarks and general suggestions:
- There are a lot of sentences which are not understandable.
- It is written several times that TGA or, especially, MaTGA8 is a member of the D subfamily of the basic region-leucine zippers (bZIP) family. At the same time, the TGA subfamilies are not showed or described in the text or on the Figs 1 and 2.
- In Abstract, it is stated: „Under the induction of salicylic acid (SA), MaTGA8 was down-regulated, while different members of the MaNPR1 family responded significantly differently. Among them, MaNPR11 and MaNPR3 showed an overall upward trend, and the expression level of MaNPR4, MaNPR8 and MaNPR13 was higher than other members.” There is no data in the manuscript about the expression of different MaNPR genes.
- It is written, that for statistical analysis Student's t-test was performed on the data, in this case use only stars (one, two or three, depending on the level of probability), not letters.
- Check and uniform the written form of scientific names of organisms (italics), abbreviated proteins (capital lettering), genes (capital letters and italics), and mutants (small letters and italics). Even the Foc tr4 is written in several forms.
Round 2
Reviewer 2 Report
The authors made significant improvement ont he manuscript, but I still found some point to correct:
page 2, line 97: phylogenetic tree - use a capital letter at the beginning
page 5 , line 155: MaNPR8 – italic
page 12, line 419: Conclusion – put into new paragraph
